# Metabolic Inflexibility as a Pathogenic Basis for Atrial Fibrillation

**DOI:** 10.3390/ijms23158291

**Published:** 2022-07-27

**Authors:** Xinghua Qin, Yudi Zhang, Qiangsun Zheng

**Affiliations:** 1Xi’an Key Laboratory of Special Medicine and Health Engineering, School of Life Sciences, Northwestern Polytechnical University, Xi’an 710072, China; xinghuaqin@nwpu.edu.cn; 2Department of Cardiology, The Second Affiliated Hospital of Xi’an Jiaotong University, Xi’an 710004, China; zooinbottle@stu.xjtu.edu.cn

**Keywords:** atrial fibrillation, metabolic flexibility, insulin resistance, Randle cycle, Warburg effect

## Abstract

Atrial fibrillation (AF), the most common sustained arrhythmia, is closely intertwined with metabolic abnormalities. Recently, a metabolic paradox in AF pathogenesis has been suggested: under different forms of pathogenesis, the metabolic balance shifts either towards (e.g., obesity and diabetes) or away from (e.g., aging, heart failure, and hypertension) fatty acid oxidation, yet they all increase the risk of AF. This has raised the urgent need for a general consensus regarding the metabolic changes that predispose patients to AF. “Metabolic flexibility” aptly describes switches between substrates (fatty acids, glucose, amino acids, and ketones) in response to various energy stresses depending on availability and requirements. AF, characterized by irregular high-frequency excitation and the contraction of the atria, is an energy challenge and triggers a metabolic switch from preferential fatty acid utilization to glucose metabolism to increase the efficiency of ATP produced in relation to oxygen consumed. Therefore, the heart needs metabolic flexibility. In this review, we will briefly discuss (1) the current understanding of cardiac metabolic flexibility with an emphasis on the specificity of atrial metabolic characteristics; (2) metabolic heterogeneity among AF pathogenesis and metabolic inflexibility as a common pathological basis for AF; and (3) the substrate-metabolism mechanism underlying metabolic inflexibility in AF pathogenesis.

## 1. Introduction

Atrial fibrillation (AF), the most commonly encountered arrhythmia in clinical practice, is characterized by irregular contractions of atrial cardiomyocytes. Driven by the increasing burden of cardio-metabolic risks (e.g., aging, obesity, and hypertension), AF prevalence has doubled over the past three decades [1], and now affects 59.7 million people worldwide [2]. AF causes substantial disability and morbidity with a high risk of heart failure (HF) [3] and ischemic stroke [4], and has exerted a tremendous burden on society, the health care system, and the economy.

To stem the expanding AF pandemic, a comprehensive AF care strategy has been proposed, including stroke prevention, rate control, rhythm control, and comorbidity/cardiovascular risk factor management. Notably, in the past 30 years, in spite of significant progress in first-line anti-AF therapies including catheter ablation and anti-arrhythmic drugs, AF is still prone to a poor prognosis even in optimally treated AF patients [5]. Lately, risk factor management (the ‘fourth pillar’) has attracted more and more attention, and has been proved to be effective in AF management by a series of clinical and experimental studies (reviewed in [6]). A better understanding of AF risk factors and the basic mechanisms underlying AF pathogenesis may shed light on the optimization of anti-AF strategy.

AF is being increasingly acknowledged as a metabolic cardiac disease. However, AF risk factors exhibit metabolic heterogeneity, and a metabolic paradox in AF pathogenesis can be seen: all AF risk factors contribute to the development of metabolic disorders, either towards (e.g., obesity, and diabetes) or away from (e.g., aging, HF, and hypertension) fatty acids’ (FAs) utilization, yet they all increase the risk of AF. Therefore, a general consensus regarding metabolic changes that predispose to AF is urgently needed. 

Metabolic flexibility is a novel concept that aptly describes switches in substrate metabolism depending on availability and requirements, thus coping with the dramatic fluctuations in energy supply and demand under physiological and pathological stimuli. Metabolic flexibility is critical for normal heart function, as it provides sufficient energy when the rapid and irregular contraction of atrial cardiomyocytes occurs during AF. Metabolic AF stressors, including aging, obesity, diabetes, hypertension, unhealthy lifestyle habits, and AF itself, all exhibit the altered availability and distorted sensing of nutrients, blunted substrate switching, and imbalanced energy homeostasis [7], implicating metabolic inflexibility as an integral component of AF pathogenesis. Since the heart is an organ with high energy consumption, cardiac metabolic inflexibility in terms of substrate sensing, trafficking, utilization and storage can significantly compromise electrical activity, pump function and anatomy [7,8], thereby inducing the electrical and structural remodeling that begets AF. 

This review article will include a brief look at the specificity of atrial metabolic characteristics, discuss the metabolic heterogeneity among AF risk factors, and propose metabolic inflexibility as a pathogenic basis for AF; recapitulate finally the cardiac substrate–metabolism mechanism underlying metabolic inflexibility and AF pathogenesis; and discuss corresponding anti-AF strategies targeting metabolic flexibility.

## 2. Metabolic Flexibility in the Normal Heart

The concept of metabolic flexibility was first proposed by Kelley et al., who found that the skeletal muscle of lean individuals showed a rapid fuel switch in response to fasting and insulin infusion compared with that of obese individuals [9]. The concept of metabolic flexibility was initially linked to the fuel selection in response to nutritional changes such as fasting/feeding transition, or insulin stimulation. Now, this concept has been expanded to the fuel selection of any given system (whole-body, organ, cell or organelle) in response to any energy stress (exercise, caloric restriction, caloric overload, hibernation, or cold exposure) [7].

The heart needs metabolic flexibility, which allows the utilization of different substrates (FAs, carbohydrates, amino acid and ketones) to maintain the contractile function in response to energy stress. Within the heart, metabolic flexibility is governed by (1) substrate availability, and (2) a complex regulatory metabolic mechanism. 

### 2.1. Substrate Availability

The heart is a muscular pump with high efficiency and rhythmicity that incessantly provides arterial blood to meet the nutritional requirements of all cells of the whole body [10]. The heart can form approximately 15–20-fold of adenosine triphosphate (ATP) of its own weight every day to meet the immense amount of energy required, the majority of which fuels contraction/relaxation (60–70% of total energy demand) while the remaining fuels ion pumps (30–40% of total energy demand) [10,11]. Regarding the high energy demands of the heart, continuous and rapid energy replenishment is a prerequisite for normal cardiac function [12].

The maintenance of substrate metabolic homeostasis is the determinant of cardiac metabolic flexibility, and the underlying mechanism has been partly investigated in recent years (for details see reviews [12,13]). Briefly, the heart is metabolically versatile and can utilize all classes of energy substrates for ATP production including FAs, carbohydrates (glucose and lactate), amino acids, and ketones. Cardiomyocytes can intake these energy substrates directly from the bloodstream or mobilize fuels from endogenous pools (triacylglycerol and glycogen). Next, substrate catabolism converges on acetyl coenzyme A (acetyl CoA) formation to fuel mitochondria through the tricarboxylic acid (TCA) cycle and subsequent oxidative phosphorylation, which operates as the major source of cardiac ATP (>95%) under non-ischemic conditions. Glycolysis (GL) in the cytoplasm and GTP generated by the TCA cycle provide the remaining energy consumed by the heart [14]. 

Cardiac fuel selection is largely determined by the type and amount of available fuel. In a healthy adult heart, FAs are the predominant substrate accounting for ~60–90% of the ATP supply [15]; glucose (glycolysis and glucose oxidation) accounts for 10–40%, and ketones contribute up to 5%. However, under pathological conditions, an apparent metabolic switch in substrate usage occurs. For example, glycolysis only provides 5–10% of the overall ATP production, whereas it increases by as much as 10- to 20-fold under hypoxic conditions such as hypoxia, anoxia or ischemia [13]. In addition, ketones are of low concentration in plasma under physiological conditions [16]; however, after overnight fasting, ketones provide 10–20% of the overall energy supply, providing a compensatory source of energy in nutritionally deficient states such as HF, ketosis, and fasting/starvation [17,18].

### 2.2. Metabolic Regulatory Network

Oxygen/substrate, total phosphates (ATP, ADP, and AMP) and phosphocreatine, inorganic phosphate, calcium, metabolites related to redox state and phospho-transfer systems have all been identified as key endogenous signaling molecules and can dynamically change in response to mechanical load and the metabolic environment of the heart. These signaling molecules were highly regulated by multiple metabolic sensors, such as AMP-activated protein kinase (AMPK) [19], peroxisome proliferator-activated receptor (PPAR) [20,21], protein kinase B (Akt) [22], hypoxia-inducible factor-α (HIF-1α) [23,24], and peroxisome proliferator-activated receptor γ coactivators 1α (PGC-1α). These stressors can orchestrate the cellular response to such signals and perform opposing, complementary or interlinked functions in the cardiac metabolism and multiple biological actions at transcriptional, post-transcriptional, and allosteric levels.

## 3. Metabolic Flexibility Matters in AF Burst

### 3.1. Specificity of Atrial Metabolic Characteristics

The heart is heterogeneous at the macroscopic, cellular and genetic levels (summarized in Figure 1). Unlike the ventricle, the atrium is relatively small and has a thin-walled structure. The atrium tends to dilate with increasing pressure, showing a positive relationship between atrial volume and pressure, which is absent in the left ventricle [25]. In cellular levels, atrial tissue contains more abundant fibroblasts (24.3% vs. 15.5%) and immune cells (10.4% vs. 5.3%), and less cardiomyocyte (49.2% vs. 30.1%) relative to the ventricular region [26]. In concert, the atrium shows greater fibrotic responses than the ventricle due to the specificity in the atrial structure and cell population [27]. As a result, the atria exhibit a 20-fold greater fibrosis than the ventricles in experimental chronic HF [28]. Substantial differences in gene expression between atrial and ventricular tissues were found among species [29]. It has been reported that the genes preferentially expressed in the atrium (atrial-predominant genes) are involved in signal transduction, cell–cell communication, fibrosis and cell apoptosis, whereas the ventricle shows a stronger expression of genes related to metabolism (ventricular-predominant genes) [27,29,30]. However, under certain pathological conditions such as AF, the number of differences in gene expression between the atrium and ventricle may decrease, and a down-regulation of atrial-predominant genes and an up-regulation of ventricular-predominant genes, including metabolism-related genes, have been observed [31].

The atrium exhibits a unique metabolic feature distinct from the ventricle. The most salient difference between the atrium and ventricle associated with metabolic inflexibility is the low metabolic reserve in the atrium. Metabolomics analysis has demonstrated that the levels of high-energy phosphate (e.g., ATP, ADP, and AMP), acetyl CoA, and metabolites in the TCA cycle (e.g., succinate, fumarate, and malate) are higher in the ventricles than in the atria, indicating lower metabolic activities in the atria [32]. In concert, high-energy phosphate levels in the atria are about one half of those in the ventricles of dogs; ATP precursor adenosine was about equal in the atrial and ventricular tissues, yet, under ischemia, the atria produced less adenosine compared with the ventricles [33]. According to a human microarray study, the transcription of genes that participate in metabolic and energy-deriving processes is less-represented in the atrium than the ventricle [30]. More importantly, the levels of key atrial enzymes involved in the complex cardiac metabolic regulatory network are relatively low in terms of activity and concentrations [34]. In a study conducted by Shimura et al., capillary electrophoresis and mass spectrometry were utilized to comprehensively measure the metabolites of atria and ventricles from male C57B6 mice at the age of 6 weeks, and it was found that the energy metabolism of the atrium mainly depends on fatty acid oxidation (FAO), while the energy metabolism of the ventricle is more flexible [35]. These results together indicate the limited atrial metabolic reserve, which leads to the low resistance to diminished nutrient/oxygen availability, rendering the atria more vulnerable to metabolic stresses. However, AF increases the metabolic reserve in the atrium by up-regulating ventricular-predominant genes (e.g., metabolic processes) [31].

Collectively, the specificity of atrial morphology and genetic phenotype may amplify the effect of divergent metabolic tolerance, and make the atrium more vulnerable to pathological electrical and structural remodeling [30]. A solid understanding of the atrial metabolic features can deepen the understanding of the metabolic inflexibility underlying AF. 

### 3.2. AF Is an Energy Stress That Uses Fuel Selection to Meet High Energy Demand

The maintenance of normal heart function requires a continuous and stable energy supply, and energy depletion by metabolic impairment, especially decreased glucose metabolism, may facilitate the initiation and progression of AF [36]. It has been reported that metabolic remodeling precedes electrophysiological, contractile and structural remodeling in AF [37]. Both transcriptomics and metabolomics studies have demonstrated that the cardiac energy metabolism of AF patients switches to a more fetal phenotype, from FAs metabolism to glycolysis [31], which could be reversed by cardiac surgery [38]. Supportively, a microarray analysis conducted by Barth reported a transcriptional down-regulation of FAs metabolism-related genes and a concomitant up-regulation of glucose metabolism-related genes in AF [31]. Key metabolic regulators behind this have been widely explored, and the LKB1-AMPK axis stands out as the most promising for AF treatment, with the finding that LKB1 knockout mice develop spontaneous AF. AMPK has pleiotropic roles, and, metabolically, it can stimulate glucose uptake and FAs metabolism, promoting metabolic flexibility and serving as a therapeutic target for treating AF [36]. Therefore, the maladaptive switch from FAs’ metabolism to glucose oxidation may provide a substrate for triggered activity and re-entry, thereby promoting AF initiation and progression.

### 3.3. Consequences of Atrial Metabolic Inflexibility

In response to developmental and environmental changes, including a variety of physiological (such as fasting/feeding, exercise, dietary pattern changes) and pathological conditions (such as obesity, diabetes, and HF), the heart must be highly metabolically flexible in the management of substrate metabolism in order to maintain cardiac function in such conditions.

It has been reported that cardiac metabolic flexibility can modulate important biological signaling pathways involved in cellular growth, survival, and death, and that it functions by modifying metabolites derived from substrate metabolism; thus, a metabolically inflexible heart may undergo profound structural remodeling [12], thereby providing a substrate for AF.

Electrical activity is a salient cardiac feature, and maintaining normal electrical conduction is crucial for heart function. Accumulating evidence has noted that impaired metabolic homeostasis can influence the excitability and impulse propagation of cardiomyocytes, as supported by the alterations in sinoatrial node function, electrical conduction, and action potential morphology in metabolically unhealthy individuals [39,40,41]. 

## 4. Metabolic heterogeneity among AF Stressors

A variety of stressors, including aging, obesity, hypertension, and diabetes, have been identified as AF risk factors [6,42]. Although these stressors all show metabolic abnormalities, they exhibit different metabolic characteristics (summarized in Table 1). These stressors are mainly divided into two major categories based on the switch towards either a pro-fatty acids oxidation (pro-FAO) or pro-glycolysis (pro-GL) metabolic phenotype. 

### 4.1. Pro-FAO Stressors: Obesity, and Diabetes

Pro-FAO stressors, including obesity and diabetes, are characterized by lipid overload and subsequent FAO acceleration at the expense of decreased glucose oxidation with/without glycolysis, and usually represent an energy-rich condition.

Obesity, the second biggest attributable risk factor for AF after hypertension [69], accounts for 17.9% of all AF cases in the Atherosclerosis Risk in Communities Study (ARIC), together with overweight [64]. A meta-analysis of 51 studies including 626,603 subjects demonstrated a 10–29% increased risk of incident, post-operative, and post-ablation AF for a 5-Unit increase in Body Mass Index (BMI) [45]. The epidemiology of AF in obesity was reviewed in [47]. Obesity induces a metabolic shift towards FAs’ utilization and a resultant enhancement of FAO, with a concurrent decrease in glycolysis and insulin sensitivity [70]. 

Emerging evidence has shown that AF risk increases in either Type 1 Diabetes Mellitus (T1DM) or Type 2 Diabetes Mellitus (T2DM). A prospective case–control study including 36,258 patients with T1DM and 179,980 controls showed that T1DM was associated with a modest (13%) increase in AF risk in men, and a significant (50%) increase in AF risk in women [49]. T2DM was reported to increase AF risk in a meta-analysis (39% or 49%) [51,52], the ARIC Study (35%) [53] or the Women’s Health Study (37%) [54]. Likewise, FAs metabolism is enhanced in both type 1 and type 2 diabetic hearts [71]. 

### 4.2. Pro-GL Stressors: Aging, Physical Inactivity; Myocardial Infarction; Hypertension, and HF

Pro-GL stressors are further divided into two categories: the energy-undemanding condition (aging and physical inactivity) and the energy-demanding condition (myocardial infarction, and hypertension). 

The energy-undemanding condition

The energy-undemanding state is associated with lower resting metabolic rates in the elderly or individuals with lower physical inactivity. Age is the foremost risk factor for AF, affecting 2% of the total population and up to 9% of octogenarians [56]. In the Rotterdam study, AF prevalence was 5.5%, rising from 0.7% in the age group 55–59 years to 17.8% in those aged 85 years and above [57]. It has been reported that, in the aging heart, glycolysis and glucose catabolism are increased at the expense of FAO [72]. 

Physical inactivity is also a major AF contributor of increasing interest. The Cardiovascular Health Study demonstrated that 26% of new AF cases were attributable to a lack of physiological activity, and moderate intensity exercise resulted in a 28% lower risk of AF compared with no regular exercise [60]. The Nord-Trøndelag Health Study 3 (HUNT3) showed that physical activity exerted an anti-AF effect independent of obesity [61]. Likewise, a growing body of data suggests that physical inactivity is associated with increased glycolytic capacity and insulin resistance, and decreased FAO capacity [73].

The energy-demanding condition

Energy-demanding conditions include oxygen-insufficient conditions such as myocardial infarction and substrate-insufficient conditions such as hypertension and HF.

Myocardial infarction and ischemic HF increase AF risk dramatically. AF incidence in patients admitted to hospital with acute myocardial infarction varied between 6.8% and 21%. The epidemiology of AF in myocardial infarction was reviewed in [62]. In myocardial infarction or ischemic HF, cardiomyocytes were ischemic and lacked sufficient oxygen, thus resulting in the activation of glucose uptake and glycolysis to increase the efficiency of ATP produced in relation to oxygen consumed. 

Hypertension is the biggest attributable risk factor for AF, and accounted for 21.6% of all AF cases in the ARIC study [64]. Among AF patients, hypertension accounts for ≈60% to 80% of patients with established AF [65]. The epidemiology of AF in hypertension was reviewed in [66]. Hypertension was also reported to switch from FAO to glycolysis [74]. 

Heart failure is the strongest predictor of AF risk. In the Framingham Study, heart failure increased the risk of AF by 5-fold in men and 6-fold in women [63]. AF risk increased dramatically with increasing heart failure severity. The epidemiology of AF in heart failure was reviewed in [68]. Metabolic changes in the failing heart involve a metabolic shift toward a greater reliance on glycolysis and ketone oxidation, with a decrease in glucose oxidation [8]. In addition, the failing heart is also insulin-resistant [8].

Taken together, metabolic heterogeneity is apparent among AF risk factors, thus hindering the understanding of the metabolic disorder underlying pathogenesis. Therefore, a general consensus regarding the metabolic changes that predispose patients to AF is urgently needed.

### 4.3. AF Classification Based on Cluster Analysis

AF is a very heterogeneous condition caused by a variety of stressors, and its classification is usually based on disease subtype, left atria size, or thromboembolism risk. Recently, cluster analysis was performed to identify unique clinically relevant AF phonotypes [75,76,77]. A cluster analysis of the J-RHYTHM registry of AF patients classified these patients into four categories: younger/low comorbidity cluster, hypertensive cluster, high bleeding risk cluster, and atherosclerotic comorbid cluster [77]. Of note, among the four clusters, AF patients in younger/low comorbidity cluster showed considerably lower rates of risk factors and comorbidities. However, glucose tolerance and metabolic parameters, such as the quantitative insulin sensitivity check index (QUICKI), are not evaluated in these patients, thus they may have a certain degree of metabolic disorder. In addition, AF patients in this cluster showed higher alcohol consumption, which is associated with metabolic disorders and increased risk for AF [78,79].

### 4.4. Electrophysiological Properties under pro-FAO or pro-GL State

Atrial electrical remodeling is a main driving-force in the development of AF, characterized by shortening of action potential duration (APD) and effective refractory period (ERP). Accumulating evidence shows that both pro-GL or pro-FAO stressors can increase the risk of developing ECG abnormalities, including increased P-wave duration and dispersion, reduced atrial conduction velocity with increased conduction heterogeneity, and shorter ERP [80]. However, these results are not consistent. Mahajan’s and our studies showed no apparent changes in ERP in chronically obese sheep [80] and obese mice [81], whereas others showed decreased ERP [82].

Electrical remodeling is associated with changes in ion current, including an increase in transient outward current (I_to_; carried by K_V_4.2/4.3 channels), and a reduction in the L-type Ca^2+^ current (I_Ca,L_; carried by Ca_V_1.2 and Ca_V_1.3 channels) and the ultrarapid delayed rectifier K^+^ current (I_kur_; carried by K_V_1.5 channel), providing a substrate for AF. Intriguingly, K_V_1.5 expression was reduced in AF patients [83] and hypertension (pro-GL) [84], whereas its expression increased significantly in the atria of HFD mice (pro-FAO) [82,85]. 

Although we show some differences in electrophysiological properties under pro-FAO and pro-GL states, these cannot be simply attributed to the pro-FAO or pro-GL effect; especially, some results are conflicting in different studies or models. In addition, metabolic disorder may be an initiating factor in AF pathogenesis, which cannot provide sufficient energy to conquer AF stress; therefore, pro-FAO and pro-GL metabolism may have a similar effect on downstream electrical remodeling. Overall, the effects of pro-FAO and pro-GL metabolism on electrical remodeling remain to be fully verified.

## 5. Metabolic Inflexibility as the Basis of Pathogenesis among AF Stressors

Metabolic inflexibility is common in pathological conditions. It has been reported that metabolic abnormalities in adults, including obesity, insulin resistance and/or T2DM, induce a state of impaired metabolic flexibility [48]. Stull et al. showed that insulin resistance is a major contributor to metabolic flexibility in humans, and metabolic flexibility is negatively associated with aging [58]. In addition, metabolic inflexibility is also evident in the failing heart [8]. 

Therefore, we propose a negative relationship between AF risk and metabolic flexibility, as shown in Figure 2. Under physiological conditions, the heart is highly metabolically flexible, which is associated with lower AF risk, whereas under pathological conditions metabolic flexibility is impaired and causes the energy substrate preference to switch either towards FAs or glucose, resulting in increased AF risk. The underlying mechanisms of the switch in energy substrate preference contributing to metabolic inflexibility will be discussed in the following section.

## 6. The Substrate-Metabolism Mechanism Underlying Metabolic Inflexibility

### 6.1. Substrate Metabolic Flexibility 

Metabolic inflexibility has been implicated in decreased FAO capacity in FAs’ metabolism and insulin resistance and/or impaired insulin signaling in glucose metabolism. In the following, we will discuss how the alterations in each substrate metabolism (glucose, FAs, amino acids and ketones) and their regulatory signaling affect metabolic flexibility.

#### 6.1.1. Glucose Metabolic Inflexibility Underlying AF

Glucose metabolic abnormalities and pathogenesis

There is increasing evidence suggesting that abnormal glucose metabolism is crucial for AF pathogenesis. Diabetes was shown to be correlated with a 34% greater risk of developing AF in a meta-analysis [52], and to increase by 3% for each additional year of treatment [86]. Epidemic studies have shown that hyperglycemia is also associated with an increased risk of AF [86,87]. In addition, glucose fluctuations have been reported to play a key role in AF pathogenesis, as evidenced in diabetic rats [88] and human subjects with diabetes [89]. Chao et al. demonstrated an increase in the activation time of both atria and a decrease in bipolar voltage in patients with abnormal glucose metabolism, compared with those without [90]. 

Insulin resistance and AF pathogenesis

Insulin resistance, a key component of metabolic inflexibility, has been suggested as an independent risk factor for incident AF even before diabetes develops. However, some controversies regarding the role of insulin resistance in the epidemiology of AF still remain. Increasing numbers of studies have shown that metabolic syndrome, of which insulin resistance is a key component, is highly associated with AF risk. Lee reported that high values of insulin resistance assessed by homeostasis model (HOMA-IR), an insulin resistance index, were significantly associated with an increased risk of AF independent of other known risk factors in nondiabetic subjects [91]. Conversely, several studies showed neutral or even opposite results. A cohort study proposed a negative relationship between fasting plasma insulin levels and AF risk [92]. Several large cohort studies reported no independent association between insulin resistance and AF development [93,94]. 

However, emerging animal studies have identified the positive relationship between IR and AF risk. Chan et al. showed an increase in AF susceptibility in insulin-resistant rats fed high-fat and high-fructose/cholesterol diets for 15 weeks [95]. In addition, the loss of insulin signaling contributed to increased AF risk in Type 1 diabetic mice, and insulin treatment reduced AF susceptibility [96,97].

Mechanisms of glucose metabolic inflexibility underlying AF

Glucose metabolic flexibility relies on the configuration of metabolic pathways that manage glucose availability, uptake, glycolysis and glucose oxidation. 

The pro-glycolysis effect in pathological conditions is always correlated to impaired insulin sensitivity. Insulin resistance is connected with abnormalities in switching between lipid and glucose utilization, thus acting as a key component of metabolic inflexibility. The potential mechanism underlying insulin-resistance-induced AF is associated with the decreased expression of Na_V_1.5 and sodium current (I_Na_) [96], the abnormal up-regulation of calcium-homeostasis-related proteins (CaMKII) [95], and the decreased expression and membrane translocation of glucose transporter type 4 (GLUT4) and GLUT8 [98] in an insulin-resistant state. 

It was reported that glucose uptake and glycolysis are markedly increased in the failing heart, whereas glucose oxidation and mitochondrial function are decreased, indicating the uncoupling of glycolysis and glucose oxidation [8]. This phenomenon is much like the “Warburg effect” commonly observed in rapidly growing tumor cells. Liu et al. reviewed and confirmed the existence of the Warburg effect in AF [99]. The signaling pathway involved in the Warburg effect during AF includes the anti-Warburg-effect AMPK and pro-Warburg-effect pyruvate dehydrogenase kinase (PDK) and HIF-1α. 

Unlike AMPK, the negative regulator of the Warburg effect, which is widely accepted to promote metabolic flexibility and decrease AF risk, pro-Warburg PDK and HIF-1α, is underestimated in their role in AF pathogenesis.

PDK is a key regulator of glycolysis–GO coupling by phosphorylating and inactivating the pyruvate dehydrogenase (PDH), leading to increased glycolysis and FAs’ metabolism. The heart needs to oxidize enough carbohydrate to meet energy needs, and previous reports showed that PDK overexpression can cause a loss of metabolic flexibility and exacerbate cardiomyopathy [100]. In concert, the genetic inactivation of PDK4 improves hyperglycemia and insulin resistance [101]. AF is associated with stimulated PDK expression. The specific inhibition of PDK4 via Dichloroacetic acid (DCA) can attenuate metabolic stress, myocardial fibrosis remodeling, and atrial intracardiac waveform activity in a paroxysmal canine model of AF [102]. 

HIF-1α is a transcription factor driving the transcription of a variety of glycolysis-related genes, including PDK, GLUT1, hexokinase II (HKII), and lactate dehydrogenase A (LDHA), thus serving as a key regulator of glycolysis. 

It has been reported that the ratio between glycolytic and oxidative enzyme activities is correlated negatively with insulin sensitivity [103], indicating the pivotal role of glycolysis in metabolic flexibility. However, the role of the Warburg effect in AF pathogenesis has been under-estimated.

#### 6.1.2. FAs Metabolic Inflexibility Underlying AF Pathogenesis

FAO and AF pathogenesis

FAs are the preferred substrate of the heart, especially the atria. Excessive FA availability is highly relevant to AF. An increased circulating level of FAs is found in multiple metabolic AF etiologies, including obesity, T2DM [104], HF [105], and aging [106]. In a prospective cohort study, plasma phospholipid 16:0, the most abundant saturated FA in diet and circulation, was positively associated with incident AF [107]. In another observational study, high levels of circulating ceramides and sphingomyelins with palmitic acid (C16:0) appeared to increase the risk of incident AF [108]. A recent metagenomic sequencing and metabolomics study demonstrated higher levels of plasma palmitic acid and oleic acid (C18:0) in AF patients relative to healthy individuals [104]. 

Mechanisms of FAs’ metabolic inflexibility underlying AF

FAs’ metabolic flexibility relies on the configuration of the metabolic pathways that manage FA availability, uptake, transportation and oxidation.

Accumulating studies have attested that disrupted FA metabolism induces atrial metabolic inflexibility in the complex pathophysiology of AF. Atrial FAs’ metabolic inflexibility was further explained as the permanent augment of CD36 to the sarcolemma that leads to chronic FA overloading, as occurs in a variety of AF etiologies, such as IR, obesity [109], rapid atrial pacing [110], and aging [111]. Besides FA uptake, decreased FA transport and oxidation also contribute to FA overload. In concert, post-operation AF patients showed a repressed atrial expression of fatty acid binding protein 3 (FABP3), indicating the impairment of cytosol FA transportation [112]. In the atria of permanent AF patients, the gene [31] and protein [113] levels of FAO-related enzymes were reduced. In addition, our results demonstrate that restoring FAO, targeting carnitine palmitoyltransferase-1B (CPT-1B) via L-carnitine (its endogenous cofactor), could attenuate obesity-induced AF [81].

Physiologically, FAO capacity is coupled with FA uptake. However, under pathological conditions, the coupling between FAs uptake and FAO is disrupted, thereby inducing lipid accumulation and subsequent atrial lipo-toxicity [114]. Lipo-toxicity is commonly associated with risk factors for AF, including IR, obesity, aging, and myocardial ischemia–reperfusion, caused by toxic lipid intermediates, including long-chain acyl-CoAs, lipid peroxides, ceramides, diacylglycerols and acyl-carnitines. Atrial lipo-toxicity elicits mitochondrial and endoplasmic reticulum dysfunctions, activates apoptotic cell death signaling, and interferes with insulin-stimulated glycogen uptake, which together engage mechanisms underlying alterations in the atrial anatomy (hypertrophy and fibrosis [114]) and electrophysiology (connexin 43 lateralization, conduction propagation impairment [115]), culminating in AF. 

The FAs metabolic disorders, characterized by the uncoupling of FAs uptake and utilization, are divided into two classes: lipid accumulation and resultant lipo-toxicity, or over-activated mitochondrial β-oxidation. Of note, these two pathological conditions are both associated with impaired glucose metabolism and/or insulin signaling, as postulated by the “Randle cycle”. Lipid accumulation and lipo-toxicity aggravates insulin sensitivity and inhibits glycolysis via lipid intermediates such as ceramides and diacylglycerols [116]. Over-activated mitochondrial β-oxidation inhibits glucose oxidation via increased acetyl-CoA and NADH/NAD+ ratio, and inhibited glycolysis, targeting PK and PFK, via increasing citrate [117].

The role of diverse FAs metabolism-related enzymes and regulators in metabolic inflexibility-induced AF are still under debate, especially these involved in triglyceride turnover (diacylglycerol transferase/DAGT, acetyl-CoA carboxylase/ACC, and adipose triglyceride lipase/ATGL), FAO (malonyl-CoA decarboxylase/MCD and carnitine acetyltransferase/CrAT) and FAs delivery (FABP). In addition, little is known about the actual degree to which lipid handling is disrupted under physiological and pathological stimulations in the atrium with AF and AF risk factors.

#### 6.1.3. Amino Acids’ Metabolic Flexibility

BCAA and AF pathogenesis

Branched-chain amino acids (BCAA), including leucine, isoleucine and valine, are the only amino acids that can be utilized as a source of energy generation in the TCA. BCAA have been acknowledged as bio-energetic fuel for protein synthesis and cell growth, and as bio-active molecules regulating nutrient-sensitive pathways involved in multiple metabolic processes. 

A dysregulated BCAA metabolism confers a high degree of risk for cardiovascular diseases [118] and was recently revealed to have clinical and experimental relevance to AF. The circulating BCAA level is increased in metabolic AF stressors including IR, obesity, and HF individuals [119]. In an ongoing study (unpublished), we found that an 8-week BCAA supplement (0.75%) can significantly enhance AF invincibility and increase left atrial volume in mice. Likewise, atrial BCAA catabolic deficiency can promote angiotensin II (Ang II)-induced AF and atrial fibrosis [120], and underlie myocardial fibrosis and hypertrophy via the PI3K-AKT-mTOR pathway [121]. These results can be explained by the elevated accumulation of toxic BCAA or their derivatives (such as branched-chain α-keto acids), which induces mitochondrial oxidative stress by enhancing superoxide production, inhibiting mitochondrial complex I, and reducing superoxide dismutase (SOD) activity [122].

Mechanisms of BCAA metabolic inflexibility underlying AF

Although the specific role of BCAA catabolism in cardiac substrate metabolism is underexplored, recent studies on extra-cardiac tissues (pancreas, adipose, and skeletal muscle) have indicated that the quantity and proportion of circulating BCAA can modulate glucose, lipid, and protein handling [119]. It has been reported that elevated circulating levels of BCAA are positively associated with metabolic disorder and insulin resistance [123]. Mechanistically, (1) the BCAA-induced activation of mammalian target of rapamycin complex 1 (mTORC1) and the resultant negative feedback regulation of insulin signaling, and (2) the accumulation of toxic BCAA metabolites and the resultant mitochondrial dysfunction could be two plausible explanations linking BCAA and insulin resistance [124]. 

Restoring BCAA metabolic flexibility is an under-appreciated anti-AF metabolic strategy, and the underlying mechanism and the independent impacts of different AA components await further validation. 

#### 6.1.4. Ketones and Metabolic Flexibility

Ketones and AF pathogenesis

Ketones, namely β-hydroxybutyrate (β-OHB), acetoacetate (AcAc) and acetone, are initially synthesized in liver mitochondria through “ketogenesis” and can circulate to extrahepatic tissues for terminal oxidation. Serum β-OHB, the predominate (70%) circulating form of ketones, can be transported into cardiomyocytes via mono-carboxylate transporters (MCT1 and MCT2) or simple diffusion, then converted to acetyl-CoA via β-OHB dehydrogenase (BDH1) and succinyl-CoA:3-oxoacid-CoA-transferase (SCOT) in the mitochondrial matrix, and finally enter into the TCA cycle to generate ATP [125]. 

The heart utilizes ketones in proportion to their delivery determined by circulating content, thus generally make a minor contribution to cardiac energy supply under basal conditions (<3%) [126]. Hence, cardiac ketone metabolism is augmented in parallel to stimulated hepatic ketogenesis, as occurs in the nutrient deprivation or diminished carbohydrate availability that accompanies encompassing starvation/fasting, exercise, and ketogenic diets [127]. Interestingly, most of the AF risk factors, such as diabetes/IR [127], congestive HF [128,129], and dilated and hypertrophic cardiomyopathies [130], can stimulate hepatic ketogenesis, which is presumably associated with cardiac dysfunction, hemodynamic abnormalities and increased neuro-hormonal stress-related lipolysis. In a metabolomics profiling analysis of atria in AF patients, the concentrations of β-OHB, tyrosine, leucine (ketogenic AA) and fumarate (a metabolic intermediate in the TCA cycle) were reported to be elevated, indicating that ketone metabolism might be up-regulated in AF [131]. 

Mechanisms of ketone metabolic inflexibility underlying AF

Ketones are an alternative energy source in the energetically compromised heart; and are thus capable of promoting metabolic flexibility. Supportively, ketones can improve energy efficiency in the failing heart [130], and decrease the infarct size and myocardial cell death following ischemic injury [132]. These findings are further explained by ketones’ oxygen-efficient nature, which improves cardiac cell excitation–contraction coupling during hypoxia, as evidenced by its higher P/O ratio (2.50) relative to FAs (2.33), and the ability to inhibit FAO and accelerate the mitochondrial energy transduction of GO in working rat hearts [133]. However, the role of ketones in regulating glucose metabolism and insulin sensitivity is equivocal, and was reviewed in [134]. Generally, a short-term ketone diet improved glucose metabolism and insulin sensitivity, whereas a long-term ketone diet showed neutral or negative results [135,136]. 

More importantly, the metabolic modification targeting ketone metabolism is complicated in AF patients, since a long-term up-regulated ketone metabolism can impose potential arrhythmogenic risks. Supportively, β-OHB can prolong the action potential by blocking the I_to_ in murine ventricular cardiomyocytes, and suppress sympathetic nervous system (SNS) activity to reduce the heart rate and cardiac energy expenditure by antagonization via G protein-coupled receptor 41 (GPR41) [137]. In addition, a recent rodent study demonstrated that long-term exogenous β-OHB administration (16 weeks) can induce cardiac fibrosis and cellar apoptosis by inhibiting mitochondrial biogenesis [138]. Ketones and intermediate metabolites (such as acetyl-CoAs) are also understudied metabolic signals that regulate the provision of fuel and mitochondrial energetics [139,140]; thus, the alteration of ketone-derived signal components possibly influences the metabolic status to confer a metabolic risk of AF. 

Given recent evidence, up-regulated ketone metabolism might be an emergency anti-AF strategy by remedying metabolic flexibility, but is not applicable for long-term AF management. A deeper understanding of ketone metabolism in AF with different etiologies is urgent in the field of AF. 

### 6.2. Metabolism Regulatory Signaling and Metabolic Flexibility

Recent investigation has elucidated the fundamental role of AMPK in maintaining metabolic flexibility, which is regarded as a “cellular fuel gauge” and a “super metabolic regulator”. In response to energy stress, AMPK is activated, thereby promoting FAO via AMPK/PGC-1α signaling and stimulating glucose uptake and glycolysis via AMPK/HIF-1α. In addition, AMPK activation also improves insulin sensitivity [141]. These results together demonstrate AMPK as a key regulator of metabolic flexibility. 

The inactivation of AMPK is highly relevant to AF, as evidenced by the increase in AF vulnerability in various pre-clinical AF models and in genetically modified strains (Cardiac LKB1 KO mice) [142]. The pharmacological activation of AMPK and its downstream regulator, PGC-1α, can retard/reverse the pathological processes underlying AF in human [143,144] and pre-clinical AF models induced by rapid atrial pacing [145,146,147] and obesity [81], in addition to genetic modified rodents [148].

Besides AMPK, a number of signaling pathways, including PGC-1α, sirtuins, and FOXO, HIF-1α, PPAR, and Akt, are also implicated in the regulation of metabolic flexibility. Their roles in metabolic flexibility and the pharmaceutical approaches to improving metabolic flexibility are beyond the scope of this review; please refer to [7,149].

### 6.3. The Substrate-Metabolism Mechanism Underlying Metabolic Inflexibility and AF Pathogenesis

Under different forms of pathogenesis, the energy substrate preference shifts either towards (e.g., obesity and diabetes) or away from (e.g., aging, heart failure, and hypertension) FAO in the atria. As previously mentioned, the atria need metabolic flexibility to enable them to switch from fat to carbohydrate oxidation in response to AF and AF-induced ischemia. The overall (glucose–FAs utilization) and internal balance (glycolysis–glucose oxidation or FA uptake–FAO) of energy substrate metabolism is crucial for normal heart function (Figure 3). 

The uncoupling of glycolysis and oxidative phosphorylation is called “the Warburg effect”, and is mainly mediated via the up-regulation of PDK4 and the resultant inactivation of PDH activity. In hypoxic conditions such as aging, hypertension and HF, the metabolic balance shifts from oxidative phosphorylation to glycolysis to increase the efficiency of ATP produced in relation to oxygen consumed. It has been reported that the glycolysis rate is negatively associated with mitochondrial function and insulin sensitivity; thus, this metabolic switch to aerobic glycolysis leads to metabolic inflexibility. However, the relevant mechanism of glycolysis in insulin resistance and metabolic inflexibility warrants further investigation.

## 7. Anti-AF Strategies Targeting Metabolic Inflexibility

An AF strategy targeting metabolic inflexibility must consider the unique metabolic profile of certain AF etiologies, thus the recommendation of prevention and treatment for AF patients must be prudent and personal. To further elaborate the individualization of AF management, AF populations are divided into two categories based on their metabolic characteristics: (1) energy-rich pro-FAO condition (e.g., obesity and diabetes), and (2) energy-demanding pro-GL condition (e.g., aging, hypertension, myocardial ischemia, and HF). 

When atrial cardiomyocytes are in an energy-rich pro-FAO condition such as obesity or T2DM, the atrium is characterized by over-whelming FAs intake and insufficient FAO in the atrium. The uncoupling of FAs uptake and FAO leads to lipid accumulation and lipo-toxicity, thereby inducing insulin resistance and metabolic inflexibility via toxic lipid intermediates. In this case, strategies restoring the healthy coupling of FAU and FAO, including caloric restriction and exercise, are the best options in redressing metabolic inflexibility. This concept is supported by the significant reduction in AF burden in obese individuals who undergo weight loss [150]. 

When atrial cardiomyocytes are in an energy-starved/demanding pro-GL state such as aging, hypertension and HF, mitochondrial dysfunction occurs, and the TCA intermediate citrate can (1) inhibit the PDH activity and GO rate, and (2) block FAO by inhibiting CPT-1B through its derivation malonyl-CoA [151]. Thus, the atrium usually enhances the metabolism of glucose or other alternative substrates (KBs) at the expense of diminished FAO to compensate for energy deficiency. Previous animal and human studies have shown that the altered preference for atrial fuel, namely metabolic remodeling, is responsible for inducing reversible electrical reconfiguration and the irreversible structural reconfiguration that predispose patients to AF occurrence and maintenance [152]. Therefore, when treating this type of AF, moderate exercise, a heart-healthy diet, and pharmaceutical approaches to improve metabolic flexibility are of higher clinical relevance. For more details on the pharmaceutical approaches to improving metabolic flexibility, please refer to [7,149].

## 8. Conclusions and Future Perspectives

Metabolic flexibility protects cardiac pump function and electrical activity against physiological and pathological stresses. Accumulating evidence has underscored the fundamental role of atrial metabolic inflexibility in atrial arrhythmogenesis, as exemplified by disorders in substrate availability, intake, delivery, utilization, storage and turnover that combine to cause pathological metabolic switch, atrial-toxicity and mitochondrial dysfunction. However, the atrial-specific metabolic profile under various AF stressors has not been definitively characterized and awaits further evaluation.

The restoration of atrial metabolic flexibility is a conceptually attractive therapeutic option to retard/reverse the pathological processes underlying AF. Recent attention has been focused on the inconsistent consequences in different AF populations exposed to similar metabolic modifications, ranging from anti- to pro- arrhythmia, which may be ascribed to the heterogeneity of metabolic substrate handling among AF etiologies. The aggressive targeting of metabolic inflexibility based on the assessment of individual metabolic status and patient preferences should be the way forward for optimizing AF management. Although several pharmacotherapies and lifestyle interventions provide options for metabolic modification, little is known about the corresponding atrial reaction and their long-term safety, superiority and sustainability in different AF populations. Future studies are required to explore the full scope of the nutritional and/or pharmacological manipulation of substrate metabolism, and to validate the therapeutic value of metabolic modification in AF populations. 

## Figures and Tables

**Figure 1 ijms-23-08291-f001:**
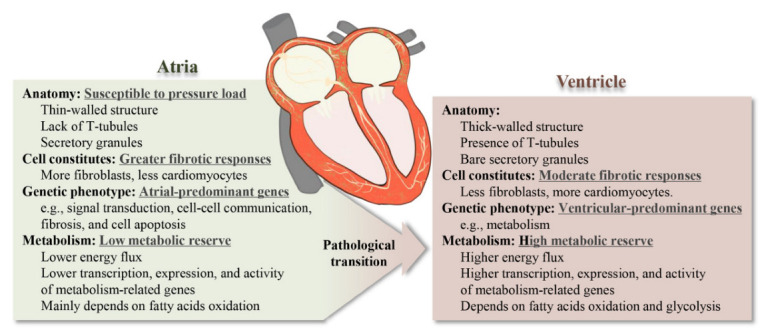
Metabolism in the healthy atrium.

**Figure 2 ijms-23-08291-f002:**
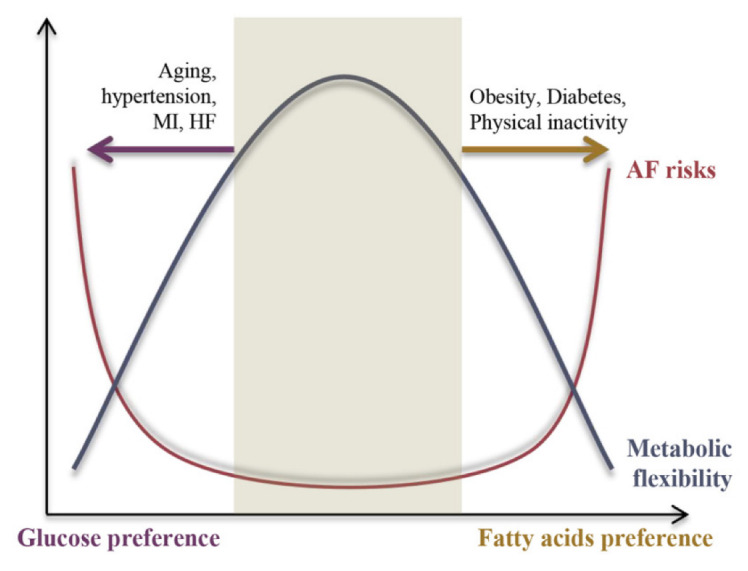
A negative correlation between the AF risk and cardiac metabolic flexibility.

**Figure 3 ijms-23-08291-f003:**
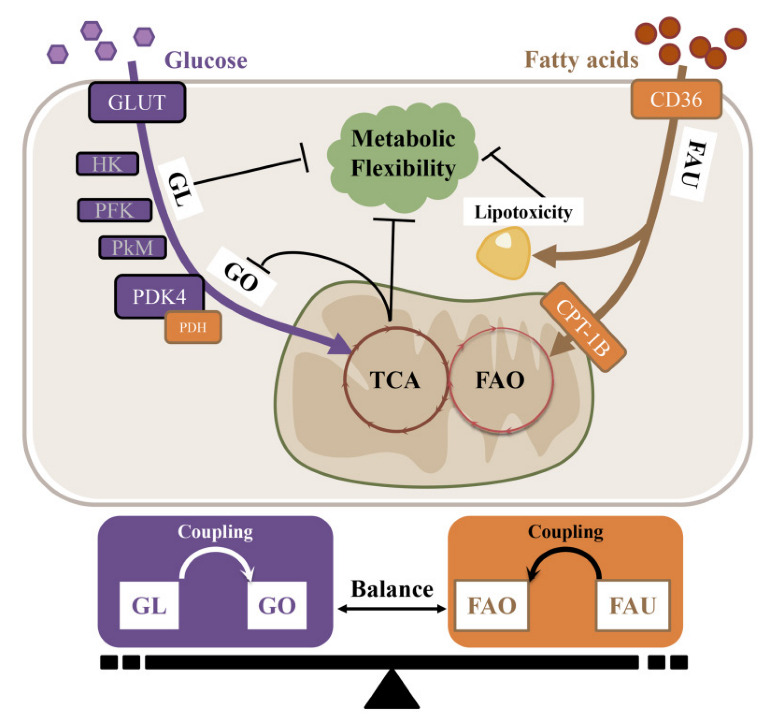
The substrate-metabolism mechanism underlying metabolic inflexibility. GL, Glycolysis; GO, Glucose oxidation; FAO, Fatty acid oxidation; FAU, Fatty acid uptake.

**Table 1 ijms-23-08291-t001:** Metabolic heterogeneity among AF stressors.

AF Risk Factors	Clinical Relevance	Metabolic Abnormalities	Pro-FAO or Pro-GL	Metabolic Inflexibility
Obesity	The Framingham cohort and the Women’s Health Study revealed a tight correlation between BMI and AF risk [43,44]; a meta-analysis of 51 studies including 626,603 subjects revealed a 20%–30% increase risk in incident AF, a 10% risk of post-operative AF, and a 13% risk in post-ablation AF for 5-Unit increase in BMI [45]; Deng et al. revealed a U-shaped relationship between obesity and post-ablation AF recurrence [46]. The epidemiology of AF in obesity was reviewed in [47]	Increased fatty acids uptake; Increased fatty acids oxidation; Increased lipotoxicity; Decreased glucose oxidation; Decreased aerobic glycolysis; Increased insulin resistance;	pro-FAO	Yes [48]
T1DM	A prospective case-control study including 36,258 patients with T1DM and 179,980 controls showed T1DM was associated with a modest (13%) increase of AF risk in men, and a significant (50%) increase of AF risk in women [49]. The epidemiology of AF in diabetes was reviewed in [50].	Increased fatty acids oxidation; Decreased glucose oxidation; Increased insulin resistance	pro-FAO	Yes
T2DM	T2DM was associated with increased AF risk in meta-analysis (39% or 49%) [51,52], the Atherosclerosis Risk in Communities Study (35%) [53] or the Women’s Health Study (37%) [54]. The epidemiology of AF in diabetes was reviewed in [50].	Increased fatty acids oxidation; Decreased glucose oxidation; Increased insulin resistance	pro-FAO	Yes [55]
Aging	It is widely accepted that aging is the most important determinant of AF risk [56]. In the Rotterdam study, AF prevalence was 5.5%, rising from 0.7% in the age group 55–59 years to 17.8% in those aged 85 years and above [57].	Increased fatty acids uptake; Decreased fatty acids oxidation; Increased glycolysis; Increased insulin resistance;	pro-GL	Yes [58]
Physical inactivity	Evidence revealed a J-shaped relationship between physical activity and AF risk [59]. The Cardiovascular Health Study demonstrated 26% of new AF cases were attributable to a lack of physiological activity, and moderate intensity exercise had a 28% lower AF risk compared with no regular exercise [60]. The Nord-Trøndelag Health Study 3 (HUNT3) showed that physical activity exerted an anti-AF effect independent of obesity [61].	Increased glycolysis; Decreased fatty acids oxidation; Increased insulin resistance;	pro-GL	Yes
Myocardial infarction	AF incidence in patients admitted to hospital with AMI varied between 6.8–21%. The epidemiology of AF in acute myocardial infarction was reviewed in [62].	Increased glycolysis; Decreased fatty acids oxidation; Increased insulin resistance;	pro-GL	Yes
Hypertension	In the Framingham Heart Study, hypertension portended an excess risk for AF by 50% in men and 40% in women [63]. In the Atherosclerosis Risk in Communities study, hypertension explained ~20% of new cases, and was the main contributor to AF burden [64]. Among AF patients, hypertension accounts for 60% to 80% of patients with established AF [65]. The epidemiology of AF in hypertension was reviewed in [66].	Increased glycolysis; Decreased glucose oxidation; Increased insulin resistance;	pro-GL	Yes [67]
HF	HF is the strongest predictor for AF risk. In the Framingham Study, HF increased AF risk 5-fold in men and 6-fold in women [63]. AF risk increased dramatically with increasing HF severity. The epidemiology of AF in HF was reviewed in [68].	Increased glycolysis; Decreased fatty acids oxidation; Increased insulin resistance;	pro-GL	Yes [8]

AF, Atrial fibrillation; FAO, Fatty acid oxidation; GL, glycolysis; HF, Heart failure; T1DM, Type 1 Diabetes Mellitus; T2DM, Type 2 Diabetes Mellitus.

## Data Availability

Not applicable.

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
