# Peer review of "Metabolic Inflexibility as a Pathogenic Basis for Atrial Fibrillation"

_ijms, 2022, doi:10.3390/ijms23158291_

Round 1

Reviewer 1 Report

This is an interesting review providing detailed analysis of metabolic mechanism in AF pathogenesis.

Before being considered for publication, this paper needs extensive English language revision.

Page 1, line 37. The authors wrote: “Notably, no significant progress has been made in the first-line anti-AF therapies including catheter ablation and anti-arrhythmic drugs over the past 30 years, and AF is still prone to progress with poorer prognosis in even optimally treated AF patient”. I think this is not correct. The most recent big trials, for example the EAST-AFNET-4, showed significant benefits in reduction of hard endpoints with the most recent first-line treatments (see for example Kirchof P, Early Rhythm-Control Therapy in Patients with Atrial Fibrillation, NEJM 2020, doi: 10.1056/NEJMoa2019422. Or Rillig A, Early Rhythm Control Therapy in Patients With Atrial Fibrillation and Heart Failure, Circulation 2021, doi: 10.1161/CIRCULATIONAHA.121.056323. I suggest to modify this sentence.

Page 3, line 147. “Metabolomics analysis has identified the contents of energy “flux”, including phosphates, acetyl CoA, and TCA cycle metabolites, were lower in the atrium than in the ventricle [32]”. This sentence is not clear, I suggesto to rephrase.

Page 3, line 149. “In concert, animal experiment has further determined the atrium can produce adenosine and inosine less under ischemia” This sentence is not clear, I suggest English language revision.

Page 4, line 157. Please wrote in extenso che meaning of acronym FAO (fatty acid oxidation?).

Page 4, line 160. “More vulnerability”. Maybe the authors intended to say “more vulnerable”?.

Page 5-7, paragraph 4. The authors correctly indicated some of the most important risk factors for AF. Of course the pathophysiology is far more complex than that, but I understand that a comprehensive clinical and pathophysiological characterization goes beyond the purpose of this paper. However, I suggest to cite this work: Malagù et al, Atrial Fibrillation in β-Thalassemia: Overview of Mechanism, Significance and Clinical Management, Biology 2022, doi: 10.3390/biology11010148.

Page 9. HOMA-IR and IR acronym should be explained.

Page 9, line 323: “gluocse”. Please correct.

Page 9, line 334: “gluocse”. Please correct.

Page 11, line 446. Please replace “is” with “are”.

Author Response

Point-by-point response to Reviewer 1

This is an interesting review providing detailed analysis of metabolic mechanism in AF pathogenesis.

Thank you very much for your careful review and constructive suggestions with regard to our manuscript, and giving us the opportunity to revise again. We have studied comments carefully, and revised the manuscript according to your comments. We hope that you will find this revised version of our manuscript acceptable for publication in International Journal of Molecular Sciences.

1. Before being considered for publication, this paper needs extensive English language revision.

Response: Thank you for your suggestions. According to your advice, our manuscript has been refined and polished by using the language editing service provided by MDPI. A certificate of language editing is uploaded along with the revised manuscript. The modified points by English editing expert are indicated in Blue.

2. Page 1, line 37. The authors wrote: “Notably, no significant progress has been made in the first-line anti-AF therapies including catheter ablation and anti-arrhythmic drugs over the past 30 years, and AF is still prone to progress with poorer prognosis in even optimally treated AF patient”. I think this is not correct. The most recent big trials, for example the EAST-AFNET-4, showed significant benefits in reduction of hard endpoints with the most recent first-line treatments (see for example Kirchof P, Early Rhythm-Control Therapy in Patients with Atrial Fibrillation, NEJM 2020, doi: 10.1056/NEJMoa2019422. Or Rillig A, Early Rhythm Control Therapy in Patients With Atrial Fibrillation and Heart Failure, Circulation 2021, doi: 10.1161/CIRCULATIONAHA.121.056323. I suggest to modify this sentence.

Response: Thank you for your valuable suggestions, and providing us detailed information on AF therapy. According to your suggestion, we have modified this sentence in the revised manuscript. Now the revised manuscript states (Page 1) that “Notably, in the past 30 years, in spite of significant progresses have been made in the first-line anti-AF therapies including catheter ablation and anti-arrhythmic drugs, AF is still prone to a poor prognosis even in optimally treated AF patients”

3. Page 3, line 147. “Metabolomics analysis has identified the contents of energy “flux”, including phosphates, acetyl CoA, and TCA cycle metabolites, were lower in the atrium than in the ventricle [32]”. This sentence is not clear, I suggesto to rephrase.

Response: Thank you for your valuable suggestions. The revised manuscript now states that “Metabolomics analysis has demonstrated that the levels of high-energy phosphate (e.g., ATP, ADP, and AMP), acetyl CoA, and metabolites in the TCA cycle (e.g., succinate, fumarate, and malate) are higher in the ventricles than in the atria, indicating lower metabolic activities in the atria”.

4. Page 3, line 149. “In concert, animal experiment has further determined the atrium can produce adenosine and inosine less under ischemia” This sentence is not clear, I suggest English language revision.

Response: Thank you for your valuable suggestions. The revised manuscript now states that “In concert, high-energy phosphate levels in the atria are about one half of those in the ventricles of dogs; ATP precursor adenosine was about equal in the atrial and ventricular tissues, yet, under ischemia, the atria produced less adenosine compared with the ventricles”.

5. Page 4, line 157. Please wrote in extenso che meaning of acronym FAO (fatty acid oxidation?).

Response: Thank you for pointing out this mistake. We have provided the full name of FAO in the revised manuscript. In addition, we have carefully checked all the acronyms thoroughly, and make sure that the full name is given at the first citation.

6. Page 4, line 160. “More vulnerability”. Maybe the authors intended to say “more vulnerable”?.

Response: Thank you for pointing out this mistake. We have corrected this error in the revised manuscript.

7. Page 5-7, paragraph 4. The authors correctly indicated some of the most important risk factors for AF. Of course the pathophysiology is far more complex than that, but I understand that a comprehensive clinical and pathophysiological characterization goes beyond the purpose of this paper. However, I suggest to cite this work: Malagù et al, Atrial Fibrillation in β-Thalassemia: Overview of Mechanism, Significance and Clinical Management, Biology 2022, doi: 10.3390/biology11010148.

Response: Thank you for your valuable suggestions. This paper has been cited in the revised manuscript, and now the manuscript states (Page 5) that “A variety of stressors, including aging, obesity, hypertension, and diabetes, have been identified as AF risk factors [6,44]”.

Reference:

44. Malagù, M.; Marchini, F.; Fiorio, A.; Sirugo, P.; Clò, S.; Mari, E.; Gamberini, M.R.; Rapezzi, C.; Bertini, M. Atrial Fibrillation in β-Thalassemia: Overview of Mechanism, Significance and Clinical Management. Biology (Basel) 2022, 11, 148, doi:10.3390/biology11010148.

8. Page 9. HOMA-IR and IR acronym should be explained.

Response: Thank you for your valuable suggestions. The full name of HOMA-IR is given in the revised manuscript, and now state that: “Lee reported that high values of homeostasis model assessment of insulin resistance (HOMA-IR), an insulin resistance index, were significantly associated with an increased risk of AF independent of other known risk factors in nondiabetic subjects” (Page 10). We used the full name of IR, insulin resistance, in the revised manuscript. In addition, we have carefully checked all the acronyms thoroughly, and make sure that the full name is given at the first citation.

9. Page 9, line 323: “gluocse”. Please correct.

Response: Thank you for pointing out this mistake. We have corrected this error in the revised manuscript.

10. Page 9, line 334: “gluocse”. Please correct.

Response: Thank you for pointing out this mistake. We have corrected this error in the revised manuscript.

11. Page 11, line 446. Please replace “is” with “are”

Response: Thank you for pointing out this mistake. We have corrected this error in the revised manuscript.

Reviewer 2 Report

Qin et al. reviewed recent information on the metabolic inflexibility in patients with pathogenesis of atrial fibrillation (AF). They clearly classified AF patients into 2 categories based on their metabolic characteristics based on the many papers: 1) energy-rich pro-fatty-acid oxidation (FAO) condition (e.g., obesity, and diabetes), and 2) energy-demanding pro-glycolysis (GL) condition induced (e.g., aging, hypertension, myocardial ischemia, and HF). This is an interesting, well-written, and relevant paper to the clinicians; however, I don't think metabolic flexibility is the only cause of AF (e.g. genetic factors).

Recent papers on AF cluster analysis show that about a quarter of AF patients have fewer comorbidities, very low CHADS score (PMC5833527, PMID: 31350002, PMC8515385). Please discuss this point in the text.

The authors elaborated on the pathogenic link between metabolic inflexibility and AF. I would like you to mention the difference in the electrophysiological properties of AF that related to the pro-FAO state and the pro-GL state.

Recently, many elderly (pro-FAO), hypertensive, obese, and diabetic patients suffer from heart failure with maintained ejection fraction (pro-GL) with atrial fibrillation. So, which are the main factors of AF, pro-FAO or pro-GL?

Author Response

Point-by-point response to Reviewer 2

Thank you very much for your careful review and constructive suggestions with regard to our manuscript, and giving us the opportunity to revise again. We have studied comments carefully, and revised the manuscript according to your comments. We hope that you will find this revised version of our manuscript acceptable for publication in International Journal of Molecular Sciences.

1. Qin et al. reviewed recent information on the metabolic inflexibility in patients with pathogenesis of atrial fibrillation (AF). They clearly classified AF patients into 2 categories based on their metabolic characteristics based on the many papers: 1) energy-rich pro-fatty-acid oxidation (FAO) condition (e.g., obesity, and diabetes), and 2) energy-demanding pro-glycolysis (GL) condition induced (e.g., aging, hypertension, myocardial ischemia, and HF). This is an interesting, well-written, and relevant paper to the clinicians; however, I don't think metabolic flexibility is the only cause of AF (e.g. genetic factors).

Response: We totally agree with you that metabolic inflexibility is not the only cause of AF. Genetic factors are strong risk factors for AF, and one supporting observation is that African Americans show lower prevalence and incidence of AF despite the higher prevalence of AF risk factors (e.g. hypertension, obesity, heart failure, diabetes, and others), known as the “racial paradox” [1]. However, AF is a very heterogeneous condition caused by a variety of underlying processes and disorders, and metabolic inflexibility may play a crucial role in AF pathogenesis in a population of similar genetic background. In addition, a comprehensive functional annotation of AF susceptibility SNPs identified by GWAS and corresponding genes demonstrated that 17.4% of AF causal genes are tightly associated with metabolic processes [2]. The effect of genetic factors on AF pathogenesis is very complicated, thus we did not discuss it in this paper.

[1]. Soliman, E.Z.; Prineas, R.J. The Paradox of Atrial Fibrillation in African Americans. J Electrocardiol 2014, 47, 804–808, doi:10.1016/j.jelectrocard.2014.07.010.

[2]. Xu, C.; Zhang, R.; Xia, Y.; Xiong, L.; Yang, W.; Wang, P. Annotation of Susceptibility SNPs Associated with Atrial Fibrillation. Aging 2020, 12, 16981–16998, doi:10.18632/aging.103615.

2. Recent papers on AF cluster analysis show that about a quarter of AF patients have fewer comorbidities, very low CHADS score (PMC5833527, PMID: 31350002, PMC8515385). Please discuss this point in the text.

Response: Thank you for your valuable suggestions. We have discussed this point in the revised manuscript, and now states (Page 8) that:

“4.3 AF classification based on cluster analysis

AF is a very heterogeneous condition caused by a variety of stressors, and its classification is usually based on disease subtype, left atria size, or thromboembolism risk. Recently, cluster analysis was performed to identify unique clinical relevant AF phonotypes [82-84]. A cluster analysis of the J-RHYTHM registry of AF patients classified these patients into 4 categories: younger/low comorbidity cluster, hypertensive cluster, high bleeding risk cluster, and atherosclerotic comorbid cluster [84]. Of note, among the 4 clusters, AF patients in younger/low comorbidity cluster showed considerably lower rates of risk factors and comorbidities. However, glucose tolerance and metabolic parameters, such as the quantitative insulin sensitivity check index (QUICKI), are not evaluated in these patients, thus they may have a certain degree of metabolic disorder. In addition, AF patients in this cluster showed higher alcohol consumption, which is associated with metabolic disorders and increased risk for AF [85,86].”

References:

[82]. Inohara, T.; Shrader, P.; Pieper, K.; Blanco, R.G.; Thomas, L.; Singer, D.E.; Freeman, J.V.; Allen, L.A.; Fonarow, G.C.; Gersh, B.; et al. Association of of Atrial Fibrillation Clinical Phenotypes With Treatment Patterns and Outcomes: A Multicenter Registry Study. JAMA Cardiol 2018, 3, 54–63, doi:10.1001/jamacardio.2017.4665.

[83]. Inohara, T.; Piccini, J.P.; Mahaffey, K.W.; Kimura, T.; Katsumata, Y.; Tanimoto, K.; Inagawa, K.; Ikemura, N.; Ueda, I.; Fukuda, K.; et al. A Cluster Analysis of the Japanese Multicenter Outpatient Registry of Patients With Atrial Fibrillation. Am J Cardiol 2019, 124, 871–878, doi:10.1016/j.amjcard.2019.05.071.

[84]. Watanabe, E.; Inoue, H.; Atarashi, H.; Okumura, K.; Yamashita, T.; Kodani, E.; Kiyono, K.; Origasa, H.; J-RHYTHM Registry Investigators Clinical Phenotypes of Patients with Non-Valvular Atrial Fibrillation as Defined by a Cluster Analysis: A Report from the J-RHYTHM Registry. Int J Cardiol Heart Vasc 2021, 37, 100885, doi:10.1016/j.ijcha.2021.100885.

[85]. Fujita, N.; Takei, Y. Alcohol Consumption and Metabolic Syndrome: Alcohol and Metabolic Syndrome. Hepatol Res 2011, 41, 287–295, doi:10.1111/j.1872-034X.2011.00787.x.

[86]. Voskoboinik, A.; Kalman, J.M.; De Silva, A.; Nicholls, T.; Costello, B.; Nanayakkara, S.; Prabhu, S.; Stub, D.; Azzopardi, S.; Vizi, D.; et al. Alcohol Abstinence in Drinkers with Atrial Fibrillation. N Engl J Med 2020, 382, 20–28, doi:10.1056/NEJMoa1817591.

3. The authors elaborated on the pathogenic link between metabolic inflexibility and AF. I would like you to mention the difference in the electrophysiological properties of AF that related to the pro-FAO state and the pro-GL state.

Response: Thank you for your valuable suggestions. We have discussed this point in the revised manuscript, and now states (Page 8-9) that:

“4.4 Electrophysiological properties under pro-FAO or pro-GL state

Atrial electrical remodeling is a main driving-force in the development of AF, characterized by shortening of action potential duration (APD) and effective refractory period (ERP). Accumulating evidence shows that both pro-GL or pro-FAO stressors can increase the risk of developing ECG abnormalities, including increased P-wave duration and dispersion, reduced atrial conduction velocity with increased conduction heterogeneity, and shorter ERP [87]. However, these results are not consistent. Mahajan’s and our studies showed no apparent changes in ERP in chronically obese sheep [87] and obese mice [88], whereas others showed decreased ERP [89].

Electrical remodeling is associated with changes in ion current, including an increase in transient outward current (Ito; carried by KV4.2/4.3 channels), and a reduction in the L-type Ca2+ current (ICa,L; carried by CaV1.2 and CaV1.3 channels) and the ultrarapid delayed rectifier K+ current (Ikur; carried by KV1.5 channel), providing a substrate for AF. Intriguingly, KV1.5 expression was reduced in AF patients [90] and hypertension (pro-GL) [91], whereas its expression increased significantly in the atria of HFD mice (pro-FAO) [89,92].

Although we show some differences in electrophysiological properties under pro-FAO and pro-GL states, they can’t be simply attributed to the pro-FAO or pro-GL effect; especially, some results are conflicting in different studies or models. In addition, metabolic disorder may be an initiating factor in AF pathogenesis, which can’t provide sufficient energy to conquer AF stress; therefore, pro-FAO and pro-GL metabolism may have a similar effect on downstream electrical remodeling. Overall, the effects of pro-FAO and pro-GL metabolism on electrical remodeling remain to be fully verified.”

References:

[87]. Mahajan, R.; Lau, D.H.; Brooks, A.G.; Shipp, N.J.; Manavis, J.; Wood, J.P.M.; Finnie, J.W.; Samuel, C.S.; Royce, S.G.; Twomey, D.J.; et al. Electrophysiological, Electroanatomical, and Structural Remodeling of the Atria as Consequences of Sustained Obesity. J Am Coll Cardiol 2015, 66, 1–11, doi:10.1016/j.jacc.2015.04.058.

[88]. Zhang, Y.; Fu, Y.; Jiang, T.; Liu, B.; Sun, H.; Zhang, Y.; Fan, B.; Li, X.; Qin, X.; Zheng, Q. Enhancing Fatty Acids Oxidation via L-Carnitine Attenuates Obesity-Related Atrial Fibrillation and Structural Remodeling by Activating AMPK Signaling and Alleviating Cardiac Lipotoxicity. Front Pharmacol 2021, 12, 771940, doi:10.3389/fphar.2021.771940.

[89]. Zhang, F.; Hartnett, S.; Sample, A.; Schnack, S.; Li, Y. High Fat Diet Induced Alterations of Atrial Electrical Activities in Mice. Am J Cardiovasc Dis 2016, 6, 1–9,

[90]. Van Wagoner, D.R.; Pond, A.L.; McCarthy, P.M.; Trimmer, J.S.; Nerbonne, J.M. Outward K + Current Densities and Kv1.5 Expression Are Reduced in Chronic Human Atrial Fibrillation. Circ Res 1997, 80, 772–781, doi:10.1161/01.RES.80.6.772.

[91]. Martinez‐Nunez, R.T.; Aaronson, P.I. K V 1.5 Channel Down‐regulation in Pulmonary Hypertension Is Nothing Short of MiR‐1‐aculous! J Physiol 2019, 597, 989–990, doi:10.1113/JP276390.

[92]. Aromolaran, A.S.; Boutjdir, M. Cardiac Ion Channel Regulation in Obesity and the Metabolic Syndrome: Relevance to Long QT Syndrome and Atrial Fibrillation. Front. Physiol. 2017, 8, 431, doi:10.3389/fphys.2017.00431.

4. Recently, many elderly, hypertensive (pro-GL), obese, and diabetic patients suffer from heart failure with maintained ejection fraction (pro-FAO) with atrial fibrillation. So, which are the main factors of AF, pro-FAO or pro-GL?

Response: This is a good question. We think it's a combined effect. As these patients usually show mitochondrial dysfunction and impaired oxidative phosphorylation, the cardiomyocytes switch to anaerobic glycolysis to meet the metabolic demand, which is less efficient than oxidative phosphorylation for generating ATP. Of note, obesity and diabetes increase myocardial fatty acids uptake, yet FAO is inhibited due to impaired oxidative phosphorylation. The uncoupling of fatty acids uptake and oxidation resulted in increased lipotoxicity, decreased insulin sensitivity and glucose uptake. Therefore, under this condition, the cardiomyocytes of these patients hardly provide sufficient energy in response to AF stress. We think the main factors of AF in these patients is the pro-GL effect, even it is also inhibited under this condition.